# The Yes and No of the Ethylene Involvement in Abscission

**DOI:** 10.3390/plants8060187

**Published:** 2019-06-25

**Authors:** Alessandro Botton, Benedetto Ruperti

**Affiliations:** Department of Agronomy, Food, Natural resources, Animals and Environment—DAFNAE, University of Padova, Agripolis, Legnaro, 35020 Padova, Italy; benedetto.ruperti@unipd.it

**Keywords:** ethylene, abscission, abscission zone, signaling, signal propagation, models

## Abstract

Abscission has significant implications in agriculture and several efforts have been addressed by researchers to understand its regulatory steps in both model and crop species. Among the main players in abscission, ethylene has exhibited some fascinating features, in that it was shown to be involved at different stages of abscission induction and, in some cases, with interesting roles also within the abscising organ at the very early stages of the process. This review summarizes the current knowledge about the role of ethylene both at the level of the abscission zone and within the shedding organ, pointing out the missing pieces of the very complicated puzzle of the abscission process in the different species.

## 1. Introduction

Abscission is a natural mechanism evolved by plants to respond to different cues, generated by the combination of both endogenous and exogenous factors. According to the specific organ undergoing abscission and the context in which this process is needed to occur, the balance between the different factors involved may change [1]. The abscission of organs such as petals or other flower parts is largely developmentally programmed and only in part affected by external cues, whereas the abscission of organs such as young fruits that cannot be metabolically supported by the mother plant can be significantly affected by the environmental conditions, such as light availability, temperature and drought stress [2,3].

Although different triggers may lead to abscission of either a whole organ or some of its parts, there are common steps, especially at the level of the final target of the process, i.e., the abscission zones (AZs), to which almost all the most recent studies have been addressed. The abscission physiology has been divided by different authors into four main phases or stages [4,5,6,7,8]. These models involve exclusively the AZs, without any indication of where and how the primary abscission signal is generated. The commonly accepted sequence of events occurring at the AZs includes: (i) the AZ cell differentiation, (ii) the AZ acquisition of the competence to respond to the abscission signal/s, (iii) the AZ activation caused by the abscission signals and abscission execution, and (iv) the trans-differentiation of the retained portion of the AZ to build a protective layer.

Within this physiological model, the roles of ethylene and other hormones such as auxin, and the involvement of the IDA-HAE-HSL2 pathway have been, at least in part, defined mainly in Arabidopsis and are still debated by the abscission community [9], with many open questions still to be answered, especially when this model is transferred to crops.

Concerning the generation of the abscission signal outside the AZs, available studies are quite fragmentary and some of them are outdated. Moreover, current models are often species-specific and/or organ-specific and there is no model shared by the abscission community about the propagation of the abscission signal from the abscising organ to the AZs.

With the present review, we would not like to give just an overview of current literature about abscission, as there have been several review articles published about this topic in recent years. Indeed, we would rather like to provide a comprehensive, innovative and, hopefully, fascinating point of view about the most likely links between the different steps of the abscission process, under a continuous perspective and not just as watertight compartments.

## 2. The Regulatory Role of Ethylene at the Abscission Zone

Abscission is accomplished through a series of highly coordinated sequential events that have been in large part characterized in model species such as Arabidopsis and tomato. While the genetic dissection of (i) AZ differentiation is still partially uncovered and mostly ethylene independent, with the jointless MADS box gene being one of the master regulators identified so far for the specification of AZ cells [10], the subsequent events of (ii) acquisition of competence of the AZ cells to abscising signals, (iii) the induction and execution of the abscission process and (iv) the trans-differentiation of a protective layer on the AZ proximal side, involve the action of ethylene at different levels. 

Ethylene was first identified as the primary abscission inductive factor [11,12]. The acquisition of competence of the AZ cells to respond to the abscission inductive signals results from the interplay of several factors among which the balance between ethylene and auxin represents a central regulatory point. Abscising organs, such as floral organs after pollination or senescing leaves and fruits, produce ethylene that is released to the AZ cells, thereby inducing the execution of cell separation processes and organ detachment from the plant [8,9]. The regulation of auxin transport from the organ destined to abscise plays an important role in controlling the competence of AZ cells to undergo cell separation. Auxin is generally claimed to perform a negative regulation on the sensitivity of the AZ to ethylene’s action [13,14]. This notion is based on the evidence that auxin depletion precedes the induction of natural abscission and removal of the auxin source (e.g., by leaf deblading or fruitlet embrioctomy) or weakening of auxin polar transport (by use of chemical inhibitors such as NPA, 1-N-naphthylphthalamic acid, or TIBA, 2,3,5-triiodobenzoic acid), results in a significant increase of AZ sensitivity to ethylene and an acceleration of the abscission process (reviewed by Meir et al. [14]). In fact, tissue specific auxin depletion at the AZ of Arabidopsis floral organs, mediated by the expression of bacterial genes such as IAA-Lys synthetase under an AZ specific polygalacturonase (PG) promoter, results in enhanced abscission, while the increase of IAA levels by the expression of the IAA biosynthetic gene *iaaM* results in a delay of the process [15]. These data, together with evidence showing that mutants with impaired auxin influx carriers (AUX1 and AUX1-Like3, LAX3) displayed early abscission [15], support the role of auxin as a negative regulator of AZ cells sensitivity to ethylene’s action and that auxin depletion at the AZ through the regulation of its polar transport, biosynthesis and/or conjugation is a pre-requisite for increased ethylene sensitivity. Early studies have shown that ethylene, as well as carbohydrate starvation and reactive oxygen species (ROS), mediates stress induced abscission by enhancing auxin depletion at the AZ by inhibiting its polar transport [3,15,16,17,18]. In agreement with this model, more recent evidence demonstrated that leaf deblading in tomato resulted in the downregulation of several auxin transporters of the PIN and PIN-like (PILs) families [15], possibly as a consequence of auxin depletion in the ER (endoplasmic reticulum) and decreased auxin signaling in AZ cells. Xie et al. [19] have shown that application of auxin (IAA), but not of GA_3_ or BA, could block the induction of abscission in AZ of citrus fruitlets also after ovary removal, resulting in the regulation of several ethylene-, ABA- and auxin-related genes, and thus have provided further evidence on the negative role played by auxin on the process of AZ acquisition of sensitivity to abscising signals. Nevertheless, auxin has also been shown to enhance ethylene biosynthesis at the AZ [20,21,22] and auxin signaling is required for abscission to take place, since its abscission-specific inhibition results in a delay of organ shedding [15]. This evidence raises the issue that auxin may play distinct roles during the early and late phase of the induction of the abscission process, respectively, and ask for further investigation of the complex interplay with ethylene’s action.

Ethylene is not the sole regulatory player in abscission since Arabidopsis mutants impaired in components of the ethylene perception or signal transduction pathway display delayed abscission of floral organs but do not show a complete inhibition [23,24,25], leading to the conclusion that additional factors besides ethylene are involved. From this starting point, further research on Arabidopsis has led to the identification of essential regulatory components of abscission of flower organs, the Inflorescence Deficient in Abscission (IDA) peptide ligand and its targets leucine-rich repeat receptor-like kinases (RLKs), HAESA (HAE) and HAESA-like2 (HSL2) [24,26,27]. The *ida* mutants display delayed or blocked abscission while overexpression of *ida* results in enhanced abscission [28]. Genetic evidence demonstrated that the IDA–HAE–HSL2 pathway is highly conserved in plants [29] and operates upstream of the activation of Knotted1-Like Homeobox (KNOX) transcription factors, regulating the transcription of cell wall enzymes involved in the execution of the cell separation in the AZ [8,30,31,32]. *Ida* mutants, even though displaying some ethylene sensitivity, show delayed abscission also after treatment with exogenous ethylene, thus leading to the conclusion that the IDA–HAE–HSL2 pathway may control abscission in an ethylene-independent mode [24,26,27]. However, this view has been recently reconsidered on the basis of the functional characterization of *ida* hortologs from different species (e.g., such as litchi, citrus and lupinus) that appear to be ethylene inducible similarly to the Arabidopsis *ida* gene specifically regulated in AZs of floral organs [8,29,33,34]. In addition, since in the *ida* and *hae hls2* mutants some cell-wall related enzymes are still induced in a way comparable to wild type levels, it was concluded that the IDA–HAE–HSL2 complex acts downstream of ethylene action and is required for the later steps of the execution rather than for the inductive phase of the abscission process [8]. In this context, the inhibition of abscission in such mutants is interpreted to be the consequence of dehydration of the AZ cells, due to rapid desiccation of petals, leading to the failure of the proximal cells of the abscission layer to undergo hydration and elongation, an important step required to drive organ detachment mechanically. These findings, together with evidence showing that in the ethylene mutants *etr1-1* and *ein2-1* petals do not wilt and abscise maintaining the cell elongation process at the AZ similar to that of wild-type [25,35], further reinforce the conclusion that the IDA-HAE-HSL2 pathway may act downstream of ethylene for the execution of the final steps of abscission through the coordination of the cell wall disassembly and cell separation. Nevertheless, the fact that overexpression of *IDA* induces the ectopic abscission at branch points where vestigial AZs are present [27,28,29] suggests that the IDA-HAE-HSL2 may be also involved in the determination of acquisition by the AZ cells of competence to abscise through a process that may still raise questions on its ethylene dependency. The interaction between the IDA-HAE-HSL2 pathway and ethylene signaling also poses the need for clarification of its potential crosstalk with an additional regulator of abscission, *Nevershed* (*Nev*), an ADP-ribosylation factor-GTPase activating protein (ARF-GAP) involved in the regulation of vesicle trafficking [36] and in the control of subcellular localization of IDA-HAE-HSL2 complex [37,38]. Vesicle trafficking may be an important component of AZ regulation and the asymmetric distribution of cell wall degrading enzymes and of programmed cell death markers at the two sides of the AZ represents a fundamental step in both the execution (phase iii) and later development of the protective layer (phase iv). In fact, in tomato pedicel abscission, the cells of the proximal side of AZ were reported to be enriched with two polygalacturonase genes while the cells of the distal AZ side were conversely enriched for the expression of a cellulase gene and of genes hallmarks of programmed cell death [39]. Recent data have shown that the asymmetrically enhanced lignification of cells located in the distal side of the AZ is required to impede diffusion of the cell wall remodeling enzymes from the proximal AZ side to the distal one during cell separation and this process is required for the proper trans-differentiation of the cells at the proximal side into a newly formed cuticle to prevent pathogen infection during formation of the protective layer [39,40]. How these processes may be regulated by ethylene (and/or by its interaction with additional players) is the subject of active research especially in the context of the driving role played by the abscising organ.

## 3. The Role of Ethylene within the Abscising Organ

The hypothesis that the primary signal/s triggering the activation of the AZ are generated within the shedding organ represents one of the most intriguing aspects of abscission physiology. Among these signals, ethylene is certainly a top player and one of the best candidates due to its high mobility as a gas.

Besides the studies that were carried out on model plants, most of the research regarding the intra-organ generation of the abscission signal in fruit crops was performed in apple, since its fruitlet clusters represent a unique model system for studying correlatively-driven fruit drop. Abscission induced by a transient stress condition and not by a sort of “automatism”, such as when it occurs as a result of senescence (i.e., leaf) or other developmental switches (i.e., flower verticils), is much more interesting to study and apple seems a very suitable system. The availability of thinning chemicals that can specifically stimulate fruitlet abscission allowed for obtaining populations of fruitlets with different shedding potentials in open field conditions [41,42,43], thus opening the way to different analytical approaches, especially by means of the new generation omic technologies.

Transcriptomic analyses allowed the identification of the main regulatory steps occurring within the apple fruitlets [41], in both cortex and seeds, during abscission induction, not only before the effective activation of the AZ, but even before appearance of measurable differences between abscising and persisting fruits, for example in terms of growth and ethylene biosynthesis and growth.

The adoption of a hormone-like thinning chemical, i.e., 6-benzyladenine, a cytokinin, allowed for enhancing the natural stress conditions inducing the apple fruitlets to shed from the tree according to the well-known phenomenon of “physiological fruit drop” [44]. This is the process through which fruitlet abscission naturally occurs in apple, due to the insufficient amount of assimilates that the tree is able to provide just after fruit set, when a fully functional photosynthetic apparatus is still developing. In this way, the system was not biased by the action of excessively aggressive thinning chemicals that could have generated a significant “background noise” in terms of transcriptional regulation. Inter-organ competition for assimilates stored in the wood during the previous season (i.e., starch), which naturally induces physiological fruit drop, is enhanced by BA through a stimulation of shoot growth and bud break thus further limiting sugar supply to the fruitlets [44]. The small lateral fruitlets, which are weaker sinks with respect to the central one derived from the king flower, are the first to perceive this nutritional stress at the level of both the cortex and seed. Multiple interactions among hormones (mainly ABA and ethylene) and other signalling molecules (i.e., ROS, reactive oxygen species, and sugars) are immediately triggered [41,42,43]. During this phase, an increase in ethylene biosynthesis in the fruitlets can be measured already at two days after abscission induction, with a peak at six days [41,45]. The amount of ethylene produced in this phase is not comparable to a climacteric biosynthetic system and requires a laser-based photoacoustic detection to be accurately and promptly quantified. During abscission induction, several processes occur, some of which are just a sort of stress reaction and some others represent the actual pathway leading to AZ activation. For example, the synthesis of isoprene may be aimed at both ROS-detoxification and still unknown secondary signaling [42]. Moreover, gene expression data indicate that the stable levels of abscisic acid (ABA) measured in abscising fruitlets at early stages of induction of the separation process may be due to a de novo biosynthesis and that ABA activates its signal transduction pathways. Its role in abscission induction, however, seems to be secondary, again dealing with the stress reaction such as most of the responses in which ABA is involved.

After the cortex has perceived the stress condition induced by the thinning treatment, similarly to what happens naturally, it starts to produce and perceive ethylene, and, consequently, activates its signal transduction pathway. The entity of the stress is somehow “quantified” by the cortex, which continues to produce ethylene when this condition is irreversible. At this stage, the seeds also perceive the situation as being irreversible, and a block of embryo development occurs, which leads to seed isolation and abortion as testified by transcriptomic data [41]. This crucial step would cause a depolarization of auxin transport, the enhancement of abscission zone sensitivity to ethylene, and its consequent activation. According to available data, the role of ethylene in this context seems crucial, as it may be responsible for the “transfer of information” between the cortex and the seeds. Its gaseous nature makes it one of the best messengers: it is synthetized in the cortex to signal the ongoing nutritional stress to the seeds.

Ethylene measurements indicate that also the central fruitlets increase their ethylene production, although not at the same level of the lateral ones. However, they do not abscise. A possible mechanism adopted by the seeds of central fruitlets to ensure their persistence and protect them from the abscission signal has been hypothesized that involves an ethylene receptor-based defense system [43]. In the central fruitlets, the relationship between ethylene biosynthesis in the cortex and the expression of ethylene receptor genes in seeds favours the latter, and the small amount of hormone coming from the cortex is not able to saturate the receptors, thus keeping its own signaling blocked and preventing abscission. On the other hand, the high amount of ethylene produced by the cortex of the lateral fruitlets can largely saturate the receptors of the seeds, thus triggering ethylene signal transduction leading to programmed cell death (PCD), embryo abortion, and AZ activation, as outlined above. Several aspects were pointed out by Eccher et al. [43] that further highlight the elegant mechanism evolved by seeds to protect themselves from the harmful action of ethylene. Four-ethylene receptor genes encoding MdETR1, MdETR2, MdETR102, and MdETR5 were shown to be differentially expressed within the different seed’s tissues and structures, suggesting that ethylene can be progressively intercepted by the different receptors in the various cell layers while penetrating the seeds. In this way, the amount of ethylene produced by the cortex that enters the seed is progressively depleted, so that receptors at the level of the embryo are not fully saturated, keeping it protected from dangerous ethylene concentrations.

## 4. The Transmission of the Signal from the Organ to the AZ

A crucial point of the abscission process concerns the way through which an abscission signaling complex, i.e., not just a single signal but a dense interaction among different entities—with ethylene playing a pivotal role, generated inside an organ can spread out and reach an anatomically distant target AZ. Current knowledge indicates the decreasing auxin flow through the AZ as an important trigger to unlock its activation [14]. However, once the AZ is unlocked, it needs a further stimulus to be activated, i.e., ethylene, to which it has become more sensitive as auxin flow decreased. However, where does this ethylene come from? There are different possible answers, but a careful search of the abscission literature may allow to trace a working hypothesis. Ruperti et al. [46] demonstrated that a clear decreasing gradient of ethylene biosynthesis activation is present in peach fruitlets when their abscission is induced. This ethylene biosynthetic gradient spreads, with a progressively decreasing trend, from the distal fruitlet tissue, where ethylene levels are highest, through the abscission zone to the proximal non-zone of the peduncle. An important aspect to consider is that fruitlets are photosynthetic organs at this stage, and they do have stomata, so they may behave similarly to leaves. In apple, these stomata may be induced to close by the increase in ABA concentration occurring within the fruitlet during the early abscission induction [42]. Therefore, the fruitlet transpiration rate may be strongly reduced, and the diffusion of ethylene concurrently produced during this early phase can be significantly compromised, as the lenticels are not yet developed at this stage. As a consequence, gas movement can take place through the liquid phase via the only available way out: the peduncle. Once ethylene reaches the peduncle tissues, it may stimulate its own synthesis, as demonstrated in peach [46], and progressively spread up to the AZ (also throughout its immediate precursor, i.e., ACC, then converted to ethylene) with a sufficient amount able to activate this ethylene-sensitive targets (Figure 1).

## 5. Conclusions and Perspectives

The abscission community around the world has been particularly active in the last few decades and several research groups in Europe, Israel, USA and China have developed a strong tradition and scientific skills on this topic. Thanks to this diligent community, several aspects of the abscission process have been investigated and clarified. However, there is still a scarce interaction among these groups, apart from some bilateral initiatives that represent an exception. At the current stage of knowledge development, a stronger collaboration may allow not only a significant step forward on the understanding of abscission physiology, but also to save the tradition, experience and competencies of the scientist who have dedicated most of their professional life to this topic.

A few main research lines should be coordinately faced by the abscission community, through an extension of the studies to crops or other model species:
Validation of the IDA-HAE-HSL2 pathway;Validation of the role of ethylene inside the AZs;Further investigations about the role of ethylene within different types of abscising organs/parts;Validation of the hypothesis of the propagation of the abscission signal from the organ to the AZ.

Abscission is certainly a topic with relevant potential applications in horticulture, some of which have already been actually achieved. All the efforts put in the past on basic research may now give fruitful results also from the point of view of knowledge transfer and applied research, to allow for setting up innovative strategies to either prevent or selectively stimulate abscission in a number of crops that is increasing as is the interest in this topic all over the world. What we have to do now is to step up our game.

## Figures and Tables

**Figure 1 plants-08-00187-f001:**
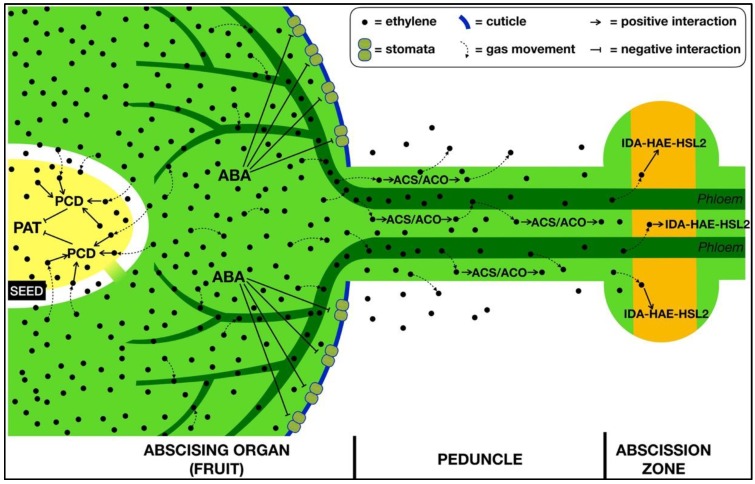
A working model for the involvement of ethylene in the whole process of abscission, taking the fruit as a reference. Upon abscission induction, ethylene and abscisic acid (ABA) are synthetized within the fruit. Ethylene enters the seed/s and block polar auxin transport (PAT) by inducing programmed cell death (PCD) of the embryo/s, so that the reduced auxin stream allows the AZ to become sensitive to ethylene. ABA stimulates stomata closure, thus preventing ethylene from exiting the fruit. The gaseous hormone is thus constricted to diffuse through the liquid phase either apoplastically or symplastically into the vascular system (i.e., the phloem), via the only available way out: the peduncle. Once ethylene reaches the peduncle tissues, it may either diffuse out from it or stimulate its own synthesis through ACS (Aminocyclopropane-1-carboxylic acid synthase or ACC synthase; EC 4.4.1.14) and ACO (Aminocyclopropane-1-carboxylic acid oxidase or ACC oxidase; EC 1.4.3.3) enzymes, so that it progressively spreads up, finally reaching the AZ with a sufficient amount able to activate the ethylene-sensitive targets, such as the IDA-HAE-HSL2 pathway. Symbols are explained in the legend.

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
