# Peer review of "The Yes and No of the Ethylene Involvement in Abscission"

_plants, 2019, doi:10.3390/plants8060187_

Round 1
Reviewer 1 Report
“The yes and no of the ethylene involvement in abscission” is a well written manuscript categorized as a review paper. However, according to the authors it is an opinion paper about the ethylene involvement in abscission. Indeed, the authors presented a brief overview of the main theme and pointed out the possible forthcoming directions of the research of ethylene during the abscission process. In several subsections, they presented the role of ethylene in the abscission zone, abscising organs and a transmission of signal from the organ to the abscission zone. They also presented a working model of ethylene action during fruit abscission.
This kind of opinion papers are worth of publishing and it makes sense to consider the suggested research lines for a cooperative research. However, a weak point of the manuscript is the selection of references. They are rather arbitrary selected and in many cases old and they do not reflect the underlined current open points in the abscission research. The integration of some recent works on abscission process would increase the significance of the manuscript as well as claims in the presented model.
Author Response
Response to Reviewer 1 Comments
Point 1: “The yes and no of the ethylene involvement in abscission” is a well written manuscript categorized as a review paper. However, according to the authors it is an opinion paper about the ethylene involvement in abscission. Indeed, the authors presented a brief overview of the main theme and pointed out the possible forthcoming directions of the research of ethylene during the abscission process. In several subsections, they presented the role of ethylene in the abscission zone, abscising organs and a transmission of signal from the organ to the abscission zone. They also presented a working model of ethylene action during fruit abscission.
This kind of opinion papers are worth of publishing and it makes sense to consider the suggested research lines for a cooperative research.
Response 1: When we submitted the manuscript, we also wrote a cover letter to the editor to explain the purpose of our manuscript and the fact that several other review articles were recently published on the same (or similar) topic. Thus we decided to give the manuscript a different character. It is still a review specifically focused on the role of ethylene in abscission, as requested, but at the end we could not avoid to summarize what we think is critical, our working model, and possible future research lines. We believe that just a cold and detached revision of the literature could have been useless.
Point 2:However, a weak point of the manuscript is the selection of references. They are rather arbitrary selected and in many cases old and they do not reflect the underlined current open points in the abscission research. The integration of some recent works on abscission process would increase the significance of the manuscript as well as claims in the presented model.
Response 2: Concerning the age of the references, in the first version of the paper there were 21 out of 42 references (50%) published in the last decade (2010-2019) five of which in the last two years, 12 out of 42 published between 2000 and 2009, and only 9 published before 2000. Indeed, we believe that a review article, which should also tell a story, must consider the milestones of its topic, so we decided to include by purpose some original and older references. That said, we welcome reviewer’s suggestions and we added some references to the manuscript, two of which are dated 2016 and 2019.

Reviewer 2 Report
Dear Authors,
The subject of this paper is of interest for the Journal and for scientists. From a wide range of specific research areas concerning the mechanisms regulating organ cutting, Authors decided to explain the spatial place of ethylene action. This issue is significant and interesting, given the fact that this hormone is involved not only in the initiation of abscission, but also takes part in the execution of this process e.g. stimulates hydrolytic enzymes. In my opinion, paper will be of interest to the international scientific community and deserves for publication in Plants, although some minor changes are recommended.
The title reflects the content of work. An abstract is concise and factual. Describes the main issues raised in the manuscript. Authors stated in lines 26-27 „…..plant can be significantly affected by the environmental conditions, such as light availability or temperature [2,3].” The quoted work of Stawicki et al. (2015) investigated also water deficit influence on organ separation, thus „drought stress” should be added. In lines 30-32 Authors written „The abscission physiology has been divided by different authors into four main phases (from phase 1 to 4) or stages (from stage A to D) [4-8]. The descriptions in parenthesis do not contain substantive content, they should be deleted because, in the next sentence, the Authors put emphasis on the individual stages of abscission.
I suggest to clarify the title of the second chapter, it is not very informative. I propose „The role of ethylene in the functioning of the abscission zone ". The crucial role of IAA-ET balance in organ shedding has been pointed out many times. However, the detailed mechanism of both hormones during abscission processes remains poorly understood. Authors pay attention to the connection between polar auxin transport and sensitivity of AZ cells to ET, but it cannot be ignored that this regulation also includes de novo biosynthesis of phytohormones in the place of action. Auxin can stimulate the production of ET and, consequently, enhance abscission in cotton and bean (Abeles and Rubinstein 1964, Morgan and Hall 1964). Moreover, Kućko et al. (2019) proved by cytochemical analyzes, that the changing level of IAA above and below flowers AZ in yellow lupine had a positive effect on ET biosynthesis genes (LlACS, LlCO) and localization of ET precursor – ACC exclusively in AZ area.
Authors stated in lines 102-104 „However, this view has been recently reconsidered on the basis of the functional characterization of ida hortologs from different species (such as litchi and citrus) that appear to be ET inducible similarly to the Arabidopsis ida gene specifically regulated in AZs of floral organs [8, 26, 30]”. Recently, Wilmowicz et al (2018) observed that ET increased expression of IDA homolog in flower AZ fragments isolated from Lupinus luteus.
Collectively, Authors raised an issue of abscission in model plants (Arabidopsis and tomato) and one crop species (apple). Giving the fact, that there are also several papers concerning phytohormonal control of abscission events in the other crops, e.g. litchi, mango, lupine, oil palm Authors should also take into account all these results. Especially in the context of global climate changes that can affect the AZ activity and yielding of many species. All these results are of most importance since can provide molecular markers for selection of varieties characterized by higher stress tolerance. Furthermore, in order to verify the hypothesis that IDA-HSL mechanism is conservative in agronomic species (Authors also stated), in my opinion, all results concerning identification and regulation of the elements of that pathway shouldn’t be omitted (suggested below).
Li C, Wang Y, Ying P, Ma W, Li J. Genome-wide digital transcript analysis ofputative fruitlet abscission related genes regulated by ethephon in litchi. Front Plant Sci. 2015; 7;6:502.
Hagemann MH, Winterhagen P, Hegele M, Wünsche JN. Ethephon induced abscission in mango: physiological fruitlet responses. Front Plant Sci. 2015; 15;6:706.
Wilmowicz E, Frankowski K, Kućko A, Świdziński M, de Dios Alché J, Nowakowska A, Kopcewicz J. The influence of abscisic acid on the ethylene biosynthesis pathway in the functioning of the flower abscission zone in Lupinus luteus. J Plant Physiol. 2016; 1;206:49-58.
Sundaresan S, Philosoph-Hadas S, Riov J, Belausov E, Kochanek B, Tucker ML, Meir S. Abscission of flowers and floral organs is closely associated with alkalization of the cytosol in abscission zone cells. J Exp Bot. 2015; 66(5):1355-68.
Taesakul P, Siriphanich J, van Doorn WG. Two abscission zones proximal to Lansium domesticum fruit: one more sensitive to exogenous ethylene than the other. Front Plant Sci. 2015;6:264. Published 2015 Apr 21. doi:10.3389/fpls.2015.00264
Roongsattham P, Morcillo F, Fooyontphanich K, Jantasuriyarat C, Tragoonrung S, Amblard P, Collin M, Mouille G, Verdeil JL, Tranbarger TJ. Cellular and pectin dynamics during abscission zone development and ripe fruit abscission of the monocot oil palm. Front Plant Sci. 2016; 26;7:540.
Please re-examine the „References” carefully because there are mistakes, e.g.: „Crosstalk between environmental stresses and plant metabolism during reproductive organ abscission”
should be „Cross-talk between environmental stresses and plant metabolism during reproductive organ abscission”
Author Response
Response to Reviewer 2 Comments
Point 1: The title reflects the content of work. An abstract is concise and factual. Describes the main issues raised in the manuscript. Authors stated in lines 26-27 „…..plant can be significantly affected by the environmental conditions, such as light availability or temperature [2,3].” The quoted work of Stawicki et al. (2015) investigated also water deficit influence on organ separation, thus „drought stress” should be added. In lines 30-32 Authors written „The abscission physiology has been divided by different authors into four main phases (from phase 1 to 4) or stages (from stage A to D) [4-8]. The descriptions in parenthesis do not contain substantive content, they should be deleted because, in the next sentence, the Authors put emphasis on the individual stages of abscission.
Response 1: Both changes suggested by the Reviewer were made.
Point 2:I suggest to clarify the title of the second chapter, it is not very informative. I propose „The role of ethylene in the functioning of the abscission zone ". The crucial role of IAA-ET balance in organ shedding has been pointed out many times. However, the detailed mechanism of both hormones during abscission processes remains poorly understood. Authors pay attention to the connection between polar auxin transport and sensitivity of AZ cells to ET, but it cannot be ignored that this regulation also includes de novo biosynthesis of phytohormones in the place of action. Auxin can stimulate the production of ET and, consequently, enhance abscission in cotton and bean (Abeles and Rubinstein 1964, Morgan and Hall 1964). Moreover, Kućko et al. (2019) proved by cytochemical analyzes, that the changing level of IAA above and below flowers AZ in yellow lupine had a positive effect on ET biosynthesis genes (LlACS, LlCO) and localization of ET precursor – ACC exclusively in AZ area.
Response 2:we agree with the reviewer’s comment on the complexity of the interplay between ethylene and auxin during abscission. We have added a brief comment on this aspect raising the interest on the fact that the auxin-ethylene cross-talk in abscission still needs further clarification. However, we did not want to comment on the auxin-induced up-regulation of ethylene genes in detail since this is a large topic already discussed at several levels (fruit ripening, Arabidopsis cell elongation, etc,). We preferred instead to briefly comment on the putative dual role played by auxin during the early and late phases of organ abscission. We hope that this decision will be acceptable for the reviewer.
Point 3:Authors stated in lines 102-104 „However, this view has been recently reconsidered on the basis of the functional characterization of ida hortologs from different species (such as litchi and citrus) that appear to be ET inducible similarly to the Arabidopsis ida gene specifically regulated in AZs of floral organs [8, 26, 30]”. Recently, Wilmowicz et al (2018) observed that ET increased expression of IDA homolog in flower AZ fragments isolated from Lupinus luteus.
Response 3:This was intended not to be an exhaustive list, however we understand the reviewer’s point and we have slightly changed the sentence and added the reference Wilmowicz et al (2018) as follows: “….ida hortologs from different species (e.g. such as litchi, citrus and lupinus) that appear to be ethylene inducible…”
Point 4:Collectively, Authors raised an issue of abscission in model plants (Arabidopsis and tomato) and one crop species (apple). Giving the fact, that there are also several papers concerning phytohormonal control of abscission events in the other crops, e.g. litchi, mango, lupine, oil palm Authors should also take into account all these results. Especially in the context of global climate changes that can affect the AZ activity and yielding of many species. All these results are of most importance since can provide molecular markers for selection of varieties characterized by higher stress tolerance. Furthermore, in order to verify the hypothesis that IDA-HSL mechanism is conservative in agronomic species (Authors also stated), in my opinion, all results concerning identification and regulation of the elements of that pathway shouldn’t be omitted (suggested below).
Response 4:We understand the reviewer’s comment however, as already mentioned, there are already reviews available dealing in detail with the precise regulatory aspects of abscission. Indeed, our aim was to provide an opinion paper highlighting the findings that we thought could be more relevant. For this reason, the list of references is not exhaustive, and we have decided to focus only on some publications. Even though the list mentioned by the reviewer is relevant, is however out of the scope of a short opinion review on ethylene’role in abscission as we have conceived it in agreement with the editor.
Point 5:Please re-examine the „References” carefully because there are mistakes, e.g.: „Crosstalk between environmental stresses and plant metabolism during reproductive organ abscission” should be „Cross-talk between environmental stresses and plant metabolism during reproductive organ abscission”
Response 5:References were carefully checked, and the suggested correction was made. New references were added as suggested.

Round 2
Reviewer 1 Report
I endorse the new version of the manuscript.